# Comparisons of Clinical Features and Outcomes of COVID-19 between Patients with Pediatric Onset Inflammatory Rheumatic Diseases and Healthy Children

**DOI:** 10.3390/jcm11082102

**Published:** 2022-04-09

**Authors:** Fatih Haslak, Sevki Erdem Varol, Aybuke Gunalp, Ozge Kaynar, Mehmet Yildiz, Amra Adrovic, Sezgin Sahin, Gulsen Kes, Ayse Ayzit-Kilinc, Beste Akdeniz, Pinar Onal, Gozde Apaydin, Deniz Aygun, Huseyin Arslan, Azer Kilic-Baskan, Evrim Hepkaya, Ozge Meral, Kenan Barut, Haluk Cezmi Cokugras, Ozgur Kasapcopur

**Affiliations:** 1Department of Pediatric Rheumatology, Cerrahpasa Medical School, Istanbul University-Cerrahpasa, Istanbul 34303, Turkey; drfatihhaslak@gmail.com (F.H.); sevkivarol@gmail.com (S.E.V.); agunalp73@hotmail.com (A.G.); ozge.kaynar2@gmail.com (O.K.); yildizmehmet@istanbul.edu.tr (M.Y.); amra.adrovic@istanbul.edu.tr (A.A.); sezgin@istanbul.edu.tr (S.S.); drkenanbarut@hotmail.com (K.B.); 2Department of Pediatric Infectious Diseases, Cerrahpasa Medical School, Istanbul University-Cerrahpasa, Istanbul 34303, Turkey; gulsen.kes.17@gmail.com (G.K.); besteak.90@gmail.com (B.A.); drpinaronal@gmail.com (P.O.); gozde_apaydin@hotmail.com (G.A.); fdenizaygun@gmail.com (D.A.); cokugras@gmail.com (H.C.C.); 3Department of Pediatric Pulmonology, Cerrahpasa Medical School, Istanbul University-Cerrahpasa, Istanbul 34303, Turkey; kilincayse203@gmail.com (A.A.-K.); huseyinarslan33@gmail.com (H.A.); kilicazer@gmail.com (A.K.-B.); dr_ekarakaya@yahoo.com (E.H.); dr.ozgemeral@gmail.com (O.M.)

**Keywords:** COVID-19, SARS-CoV-2, rheumatology, pediatrics, familial Mediterranean fever

## Abstract

(1) Background: We aimed to describe the clinical features and outcomes of coronavirus disease-2019 (COVID-19) in children and late adolescents with inflammatory rheumatic diseases (IRD) and to measure their severity risks by comparing them with healthy children. (2) Methods: Among children and late adolescents found to be severe acute respiratory syndrome-coronavirus-2 (SARS-CoV-2) positive via polymerase chain reaction (PCR) test, IRD patients with an at least six-months follow-up duration, and healthy children were included in the study. Data were obtained retrospectively. (3) Results: A total of 658 (339 (51.5%) females) (healthy children: 506, IRD patients: 152) subjects were included in the study. While 570 of 658 (86.6%) experienced COVID-19-related symptoms, only 21 (3.19%) required hospitalization with a median duration of 5 (1–30) days. Fever, dry cough, and fatigue were the most common symptoms. None of evaluated subjects died, and all recovered without any significant sequelae. The presence of any IRD was found to increase the risk of both hospitalization (OR: 5.205; 95% CI: 2.003–13.524) and symptomatic infection (OR: 2.579; 95% CI: 1.068–6.228). Furthermore, increasing age was significantly associated with symptomatic infection (OR: 1.051; 95% CI: 1.009–1.095). (4) Conclusions: Our study emphasizes that pediatric rheumatologists should monitor their patients closely for relatively poor COVID-19 outcomes.

## 1. Introduction

Coronavirus disease-2019 (COVID-19), caused by a novel virus called severe acute respiratory syndrome-coronavirus-2 (SARS-CoV-2), is responsible for an exponentially increasing number of morbidities and mortalities, globally. Although a wide clinical spectrum varies from asymptomatic disease course to life-threatening events was noted, the most common symptoms of COVID-19 are fever, nasal congestion, dry cough, myalgia, and fatigue in adults [1,2,3]. 

Since children are vulnerable in many terms, parents and pediatricians had huge concerns regarding the probable poor outcomes of COVID-19 in children in the early days of the pandemic. Fortunately, it was revealed that the disease seems to be rare in childhood, and children are likely to have asymptomatic or mild disease course [4,5,6]. The most common COVID-19 symptoms in children are cough, pharyngeal erythema, and fever, and the overall mortality rate is relatively low compared to adults [7]. Although multi-system inflammatory syndrome in children (MIS-C) caused by SARS-CoV-2 may cause devastating consequences, such as organ failure and even death, it is an extremely rare complication with an incidence of 316 per one million cases [8]. 

However, individuals with chronic health conditions were another group that made concerns rise due to their vulnerabilities. Hypertension, diabetes, obesity, chronic lung or cardiac disease, and immune deficiencies were identified as predictors of disease severity [9]. Considering hyperinflammation as the key factor in COVID-19-related tissue damage, patients with inflammatory rheumatic diseases (IRD) are challenging conundrums on this issue since their immune system is mostly disturbed due to the disease itself or long-term immunosuppressive treatment regimens [10]. 

Studies focused on IRD patients present controversial findings, probably because of limited patients and interpretation challenges due to clinical similarities between disease flares and COVID-19. Whether patients with IRD are at increased risk of severe infection remains unclear [11,12]. Furthermore, a smaller number of patients and a lack of studies in pediatric age make a risk assessment for this group harder. We aimed to describe the clinical features and outcomes of COVID-19 in children and late adolescents with IRD and to measure their severity risks by comparing them with healthy children.

## 2. Materials and Methods

### 2.1. Patients and Data Collection

Our patients with IRD were told via phone or face-to-face appointments that they must inform us immediately when they became infected with SARS-CoV-2 at the beginning of the pandemic. Their COVID-19 histories are routinely questioned in regular follow-ups, and they are allowed to call us via phone on demand. They are routinely referred to our pediatric rheumatology department when they are admitted to our hospital due to COVID-19 suspicion. Regardless of which hospital they were admitted to, their data were obtained from phone calls or face-to-face and verified by our national health registry with their permission. Then, data of all the participants were obtained from their medical records retrospectively.

Among those found to be SARS-CoV-2 positive via polymerase chain reaction (PCR) test (Bioeksen R&D Technologies Incorporated Company, Istanbul, Turkey), rheumatic patients under 21 with an at least six-months follow-up duration in our department were included as the study group. Healthy children under 18 tested in our hospital for SARS-CoV-2 and found to be positive were accepted as control group. At least one of the participants’ family members approved the written informed consent.

Subjects found to be SARS-CoV-2 PCR positive between 11 March 2020 (the date of the first COVID-19 case was seen in our country) and 11 December 2021 (the date of the first infected case by the omicron variant of SARS-CoV-2 was seen in our country) were enrolled. Patients with omicron variant of SARS-CoV-2 have not been included due to quite distinct clinical features.

According to regulations of our ministry of health, while pediatric subspecialists, such as pediatric rheumatologists, are allowed to follow their patients with chronic health conditions until they are 21, healthy individuals with no underlying diseases can be followed up by general pediatricians until they are 18. This is the main cause of the age cut-off difference between the study group and the control group in our study. 

Data of chronic health conditions, demographic variables, COVID-19 clinical features, and outcomes were obtained from their medical records, retrospectively. Although numerous IRD patients were under immunomodulatory therapy, current guidelines were followed for managing their treatment regimens when they were infected [13]. 

### 2.2. Statistical Analysis 

We performed the statistical analysis by using SPSS for Windows, version 22.0 (SPSS Inc., Chicago, IL, USA). Categorical variables were expressed as numbers and percentages. Ages, follow-up durations and lengths of hospital stay of the patients were given as the median (minimum–maximum), based on their distribution which was measured by using the Kolmogorov–Smirnov test. Categorical variables were compared by using Chi-Square test or Fisher’s Exact test, when appropriate. Ages, follow-up durations and length of hospital stay of the patients were compared using the Mann–Whitney U or Kruskal–Wallis test, when appropriate. 

We ran multivariable logistic regression analyses for assessing risk factors of symptomatic infection and hospitalization. Age, gender, receiving immunosuppressive agents, such as biological disease modifying anti-rheumatic drugs (bDMARDs) and conventional disease modifying anti-rheumatic drugs (cDMARDs), and the presence of an IRD were included in logistic regression models for both symptomatic infection and hospitalization. We stratified the patients according to their IRD types and performed a second model that included age, gender, bDMARDs, cDMARDs, and types of IRD, such as autoinflammatory diseases (AID), juvenile idiopathic arthritis (JIA), connective tissue diseases (CTD), and vasculitis for measuring risk factors of symptomatic infection. However, we were not able to do the same second model for the evaluation of the risk factors of hospitalization due to a lack of the number of hospitalized subjects. Statistical significance was defined as *p* < 0.05. Prism software (Prism 8, GraphPad Software, San Diego, CA, USA) was used to graph data.

## 3. Results

### 3.1. Study Population

Overall, 658 (339 (51.5%) females) (healthy children: 506 (76.9%), IRD patients: 152 (23.1%)) subjects with a median age of 13.06 (0.33–20.95) years were included in the study. Median follow-up of the IRD patients was 62 (6–204) months. Twenty (13.2%) IRD patients had additional non-rheumatic diseases, such as asthma (n = 7), hypertension (n = 3), gastroesophageal reflux (n = 2), inflammatory bowel disease (n = 2), attention deficiency and hyperactivity disorder (n = 2), precocious puberty (n = 1), vesicoureteral reflux (n = 1), autism (n = 1), and mitral valve insufficiency (n = 1). 

There was a certain family contact history of COVID-19 in 557 (84.7%) participants. Chest computed tomography (CT) was performed for 159 of the subjects, and suggestive signs for COVID-19 were seen in 18 of them (11.3%). While 88 of 658 (13.4%) were fully asymptomatic, the overall symptoms were: fever (n = 321, 48.8%), dry cough (n = 277, 42.1%), fatigue (n = 266, 40.4%), headache (n = 255, 38.8%), sore throat (n = 199, 30.2%), myalgia (n = 144, 21.9%), rhinorrhea (n = 136, 20.7%), nausea–vomiting (n = 66, 10%), anosmia/ageusia (n = 65, 9.9%), dyspnea (n = 64, 9.7%), arthralgia (n = 59, 9%), diarrhea (n = 57, 8.7%), anorexia (n = 53, 8.1%), abdominal pain (n = 44, 6.7%), chest pain (n = 28, 4.3%), arthritis (n = 6, 0.9%), rash (n = 6, 0.9%), back pain (n = 5, 0.8%), and conjunctivitis (n = 3, 0.5%). Twenty-one (3.2%) participants were hospitalized, and the median length of hospital stay was 5 (1–30) days. A healthy 12-year-old girl required pediatric intensive care unit (PICU) admission, and non-invasive mechanic ventilation (NIV) was performed. Besides this case, a healthy 17-year-old boy developed MIS-C in his follow-ups. Forty-eight subjects (7.3%) were prescribed antiviral, 20 antibiotic (3%), 17 hydroxychloroquine (2.6%), and 10 anticoagulant agents (1.5%) for their COVID-19 treatment. Detailed and categorized data are given in Table 1.

Forty-five of 152 IRD patients were under biologic treatment, such as canakinumab (n = 14), adalimumab (n = 13), etanercept (n = 11), anakinra (n = 2), tocilizumab (n = 2) infliximab (n = 1), rituximab (n = 1), and baricitinib (n = 1). They were defined as the biologic group, and the rest of the IRD patients were defined as the non-biologic group. Symptomatic infection frequencies in the biologic group, non-biologic group, and healthy children were 86.7%, 92.5%, and 85.4%, respectively. Hospitalization frequencies in the biologic group, non-biologic group, and healthy children were 2.2%, 9.3%, and 2%, respectively. Fatigue, fever, and headache were the most common symptoms in both the biologic (62.2%, 51.1%, 51.1%, respectively) and non-biologic group (65.4%, 61.7%, 50.5%, respectively). The most common symptoms of healthy children were fever (45.8%), cough (45.3%), and headache (35.2%). Additional data are available in Figure 1.

While fever, headache, fatigue, sore throat, rhinorrhea, nausea–vomiting, anosmia/ageusia, chest pain, abdominal pain, back pain, rash, arthritis, arthralgia, anorexia, and conjunctivitis were significantly more common in IRD patients, cough, and myalgia were significantly more common in healthy children. There was no significant difference between IRD patients and healthy children regarding the frequencies of dyspnea and diarrhea. While hospitalization was significantly more common in IRD patients (*p* = 0.003), symptomatic infection frequency was not significantly different between IRD patients and healthy children (*p* = 0.085). Detailed data are given in Figure 2.

### 3.2. Comparison of Symptomatic and Asymptomatic Subjects

The median age of the symptomatic subjects was 13.63 (0.4–20.95) years, significantly higher (*p* = 0.005) than the median age of the asymptomatic subjects (11.5 (0.33–20.47) years). Family contact history of COVID-19 was significantly more common in asymptomatic than in symptomatic subjects (100% vs. 82.3%, *p* < 0.001). Although not significant, IRD patients’ ratio was higher in the symptomatic than in the asymptomatic group (24.2% vs. 15.9%, *p* = 0.085). Additional data are given in Table 2.

### 3.3. Comparison of Hospitalized and Non-Hospitalized Subjects

IRD patients’ ratio was significantly higher in the hospitalized than in the non-hospitalized group (47.6% vs. 22.1%, *p* = 0.003). Chest computed tomography (CT) features of COVID-19 were significantly more common in the hospitalized than in the non-hospitalized group (42.9% vs. 1.4%, *p* = <0.001). Additional data are given in Table 2.

### 3.4. Risk Factor Assessment for Symptomatic Infection and Hospitalization

Increasing age (OR: 1.046; 95% CI: 1.003–1.091; *p* = 0.037) and the presence of any IRD (OR: 2.452; 95% CI: 1.014–5.929; *p* = 0.047) were found to be risk factors for symptomatic infection in model 1 multivariate logistic regression analysis. Increasing age (OR: 1.046; 95% CI: 1.003–1.092; *p* = 0.035) was significantly related with symptomatic infection in model 2, as well. 

On the other hand, the presence of an IRD was independently associated with hospitalization (OR: 5.205; 95% CI: 2.003–13.524). Detailed data are available in Table 3.

## 4. Discussion

Out of 658 (339 (51.5%) females) subjects infected by SARS-CoV-2, except for the omicron variant, 152 were IRD patients, and 506 were otherwise healthy children. While 570 of 658 (86.6%) experienced COVID-19-related symptoms, only 21 of them required hospitalization with a median duration of 5 (1–30) days. More than half of the hospitalized subjects (11/21) were IRD patients, and none but one 15.6-year-old male scleroderma patient who was receiving tocilizumab were under biological treatment. As their leading causes of hospitalization, 7 of 11 IRD patients had severe respiratory symptoms, two had feeding intolerance due to severe diarrhea and vomiting, one had persistent fever, and one had acute abdominal pain that could require surgical intervention. Fever, dry cough, fatigue, headache, sore throat, and myalgia were the most common symptoms. One previously healthy girl required PICU admission, and a boy without any underlying disease developed MIS-C in his follow-ups. No one has died, and all recovered without any significant sequelae. 

While hospitalization was significantly more frequent among IRD patients, symptomatic infection frequency was not significantly different between IRD patients and healthy children. This finding made us consider the possibility of a more cautious attitude towards IRD patients when they are infected. However, in multivariate regression analysis, the presence of any IRD was found to increase the risk of both hospitalization and symptomatic infection. Furthermore, increasing age was significantly associated with symptomatic infection. Gender, and receiving any immunosuppressive medication, such as bDMARDs or cDMARDs, was related with neither hospitalization nor symptomatic infection.

The omicron variant of the virus, which currently dominates the outbreak, causes distinct clinical findings. However, SARS-CoV-2 keeps evolving rapidly. Since the main weapon for fighting this novel foe in the early days of the pandemic was our knowledge obtained from the previous coronaviruses, there will always be numerous things to learn from the past [14,15].

Older age was closely related to severe outcomes of COVID-19 in numerous studies [2,16,17,18,19,20]. Angiotensin-converting enzyme-2 (ACE-2) was previously shown to serve to SARS-CoV-2 as an entrance gate to human cells [21]. This interaction between ACE-2 and the virus downregulates the ACE-2, decreasing an anti-inflammatory mediator called angiotensin 1–9 and relatively increasing a pro-inflammatory mediator called angiotensin II, leading to hyperinflammation [10]. Moreover, ACE-2 expression was shown to be significantly decreased by aging [22]. These findings are compatible with the idea of older infected individuals being more likely to have symptomatic infection. Correspondingly, we revealed that the increasing age was an independent risk factor for symptomatic SARS-CoV-2 infection. 

In two of the earliest pediatric cohorts, asymptomatic infection was reported in 12.9%, and 15.8% of the laboratory confirmed COVID-19 cases [4,23]. Subsequently, a multinational study was published from Europe, and 16% of the entire pediatric cohort was asymptomatic [24]. Although our hospital is a tertiary center where mostly symptomatic patients are referred, we reported a similar frequency that 14.6% of healthy children had entirely asymptomatic SARS-CoV-2 infection. 

Considering the concerns regarding the vulnerability of IRD patients to infections, studies focused on this group of patients have been conducted rapidly since the beginning of the pandemic. Given the limited number of patients, sufficient data are still not available. Firstly, adult cases were reported, and were not considered to be at increased risk of poor outcomes [25,26]. Then, pediatric case series in a broad spectrum of rheumatic diseases, such as AID and JIA, were published, with smaller sample sizes, and they draw a similar clinical picture [27,28,29]. Maritsi et al. reported that 13 of 16 children with IRD were asymptomatic [30]. It was shown in another case series from Brazil that included 14 children with IRD that 12 were under DMARD treatment and only one was hospitalized without oxygen requirement [31]. These findings made the clinicians consider that IRD patients, either adult or children, are unlikely to have a severe COVID-19 disease course.

However, with the increasing number of IRD patients infected by SARS-CoV-2, novel and distinct findings were reported. A meta-analysis showed a significantly increased risk of COVID-19 infection in adult IRD patients than in the general population [32]. Almost half of the IRD patients required hospitalization due to COVID-19 in another adult study [33]. Furthermore, severe outcomes, such as respiratory failure and mechanical ventilation requirement, were previously shown to be more common in adult IRD patients than in non-rheumatic patients [34,35].

In a single center study, while 82% of pediatric IRD patients had symptomatic infection, 18.2% were hospitalized [36]. A multi-center study reported the symptomatic SARS-CoV-2 infection frequency as 61%, and hospitalization frequency due to COVID-19 as 7.8%, in patients with childhood rheumatic diseases [37]. According to a German registry including 76 children with IRD, 58 (76%) had symptomatic infection, two (2.6%) were hospitalized, and one has died [38]. In our study, symptomatic infection and hospitalization frequencies were 90.7% (138/152), and 7.2% (11/152), respectively. 

While the most common disease was JIA in previous studies that evaluated COVID-19 patients with pediatric IRD, FMF was the most common one in our study [36,37,38]. Geographical circumstances present a reasonable explanation for this discrepancy. It is well-known that JIA is the most common rheumatic disease in childhood [39]. However, FMF is the most common periodic fever syndrome in children and our country is one of the foremost Middle Eastern countries where the highest prevalence of FMF is reported [40]. 

Although the most common disease was JIA in German registry, all those with FMF had symptomatic infection [38]. Consistently, patients with AID (the majority of them were FMF patients) have the highest frequency of symptomatic SARS-CoV-2 infection. In our cohort, more than half of reported patients were symptomatic (68/138), and 9 of 11 hospitalized patients were FMF. Therefore, we considered the FMF predominance of our study may be responsible for our relatively worse outcomes. 

However, it was unclear whether symptomatic infection was more common in children with FMF, or FMF attacks induced by viral infections may mimic COVID-19 related symptoms. Therefore, all the diagnostic clinical criteria of FMF attacks but fever (abdominal pain, chest pain, and arthritis) were excluded from the term of symptomatic infection, and data were re-analyzed [41]. Arthralgia was excluded as well as diagnostic criteria, as it often accompanies arthritis. Since fever is the most common symptom of pediatric COVID-19 patients, it was not excluded [42]. As all symptomatic subjects had extra symptoms in addition to these excluded FMF episode-related symptoms, none of the subjects’ clinical statuses changed from symptomatic into asymptomatic, and the results were entirely the same. Thus, we did not repeat them in the paper. 

Since serious infections were reported in children under biologic therapy previously, one of the most challenging concerns for pediatric rheumatologists during the pandemic was the ongoing bDMARD treatments of their patients [43]. A multi-center study from our country enrolled 113 children with IRD under a great variety of bDMARDs and did not consider biologic treatment to be related with poor COVID-19 outcomes [44]. However, given that biologic action mechanisms are quite different, they should be evaluated separately in terms of the effects on COVID-19 prognosis. There were two children with CAPS in a study describing COVID-19 patients with AID, both were under canakinumab treatment, and both had mild infection [29]. Similarly, three children with CAPS in our cohort were receiving canakinumab, had mild to moderate COVID-19 related symptoms, and none was hospitalized. Out of remaining 11 patients under canakinumab, nine were symptomatic, none had severe disease or required hospitalization. While it is proposed that the risk of COVID-19-related hospital admission increases by using rituximab, it decreases in patients under anti-tumor necrosis factor (anti-TNF) agents [32,45]. Consistently, out of 45 patients under biologic treatment, of who more than half were receiving anti-TNF agents (25/45), 39 had symptomatic infection, none were severe, but one scleroderma patient who was receiving tocilizumab was hospitalized in our study. Patients under biologic treatment were found to be at increased risk of neither hospitalization nor symptomatic infection. 

Mainly due to its retrospective manner, the study has notable limitations, as follows: (1) Since there are limited data sets describing the disease activities of the IRD patients during their COVID-19 disease courses, we could not perform the analysis; (2) we did not assess the dosages and the durations of the patients’ medications; (3) although IRD was shown to increase the hospitalization risk, disease groups could not be analyzed separately due to a low number of hospitalized IRD patients; (4) laboratory investigations and vaccination data were available for only a few; therefore, we did not include those data in this study; (5) IRD patients with a milder COVID-19 disease course may have not informed us about their PCR positivity, which may result in a selection bias. The main strength of our study is that it contains the largest cohort of COVID-19 patients with pediatric-onset IRD and is the only study that includes a healthy control group among those focused on this group of patients, to the best of our knowledge.

## 5. Conclusions

In conclusion, our study indicates an acceptable safety profile of receiving immunosuppressive medications during the pandemic and emphasizes the increased risk of both hospitalization due to COVID-19 and symptomatic SARS-CoV-2 infection in patients with childhood-onset IRD. The most significant take-home messages of the study are that pediatric rheumatologists should monitor their patients closely for relatively poor COVID-19 outcomes, and the families of children with IRD should not withdraw their medication unless they are told to. 

## Figures and Tables

**Figure 1 jcm-11-02102-f001:**
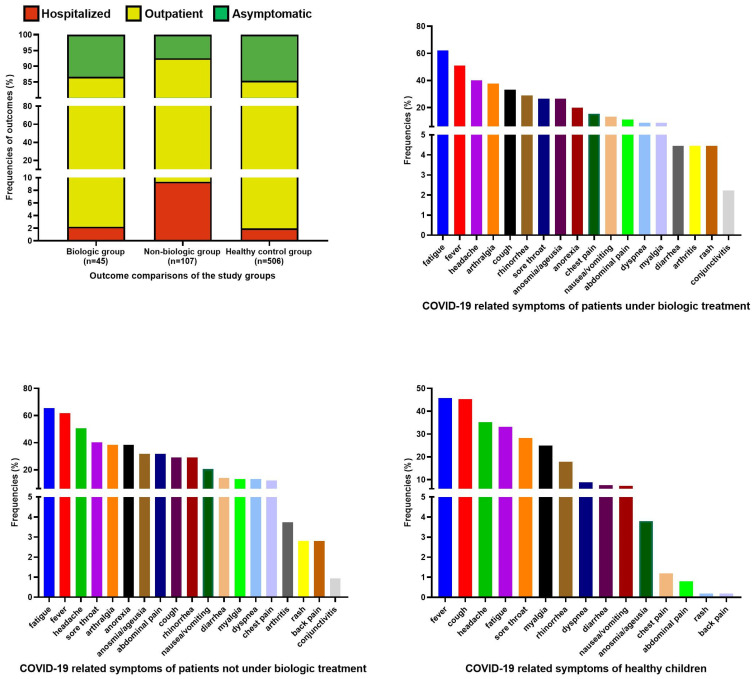
Outcome and clinical feature frequencies of the healthy control group, patients under biologic treatment, and patients not under biologic treatment.

**Figure 2 jcm-11-02102-f002:**
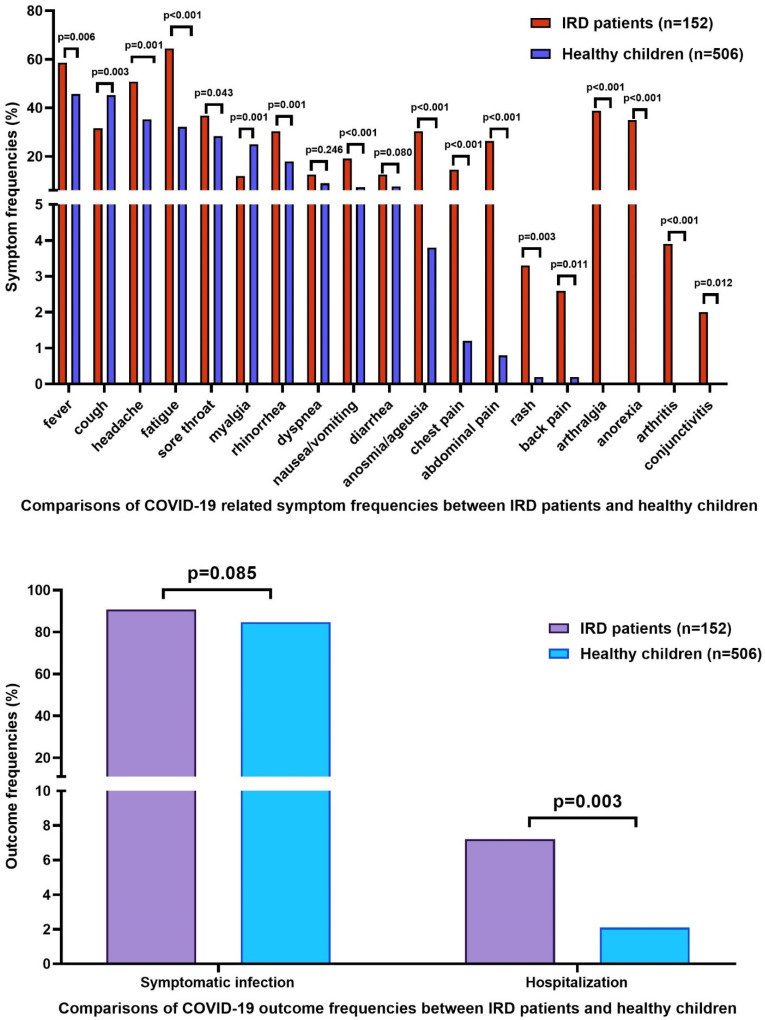
Comparisons of outcomes and clinical features of IRD patients and healthy children.

**Table 1 jcm-11-02102-t001:** Baseline characteristics of the study population.

	Healthy Control (n = 506)	Patients with AIDs (n = 81)	Patients with JIA (n = 39)	Patients with CTD (n = 22)	Patients with Vasculitis (n = 10)	*p*
Age (years) (median, min–max)	13 (0.33–17.9)	13.7 (2.94–20.86)	14.05 (3.09–20.95)	17.61 (6.8–20.59)	15.22 (3.87–20.79)	<0.001
Gender						0.283
Female (n, %)	252 (49.8%)	42 (51.9%)	24 (61.5%)	15 (68.2%)	6 (60%)	
Male (n, %)	254 (50.2%)	39 (48.1%)	15 (38.5%)	7 (31.8%)	4 (40%)	
Rheumatic diagnoses (n)	-	FMF (71) CAPS (3) PFAPA (3) HIDS (2) CRMO (1) BS (1)	oJIA (17) ERA (9) pJIA (8) sJIA (5)	SLE (10) DM (6) SD (4) Sjögren (2)	KD (3) BD (2) GPA (2) DADA2 (1) TA (1) HSP (1)	
Follow-up duration * (months) (median, min–max)	-	66 (6–182)	54 (8–192)	66 (12–170)	46.5 (16–204)	0.119
Ongoing treatments						
Colchicine (n, %)	-	75 (92.6%)	-	-	2 (10%)	
Steroid (n, %)	-	1 (1.2%)	7 (17.9%)	10 (45.5%)	3 (30%)	
bDMARDs						
Anakinra (n, %)	-	2 (2.5%)	-	-	-	
Canakinumab (n, %)	-	11 (13.6%)	3 (7.7%)	-	-	
Tocilizumab (n, %)	-	1 (1.2%)	-	1 (4.5%)	-	
Etanercept (n, %)	-	1 (1.2%)	6 (15.4%)	3 (13.6%)	1 (10%)	
Adalimumab (n, %)	-	-	12 (30.8%)	-	1 (10%)	
Infliximab (n, %)	-	-	1 (2.6%)	-	-	
Rituximab (n, %)	-	-	-	1 (4.5%)	-	
Baricitinib (n, %)	-	-	1 (2.6%)	-	-	
cDMARDs						
MTX (n, %)	-	-	14 (35.9%)	5 (22.7%)	-	
Leflunomide (n, %)	-	-	1 (2.6%)	-	-	
AZT (n, %)	-	-	-	1 (4.5%)	2 (20%)	
Cyclosporine (n, %)	-	-	1 (2.6%)	-	-	
Cyclophosphamide (n, %)	-	-	-	1 (4.5%)	-	
HCQ (n, %)	-	-	-	14 (63.6%)	-	
MMF (n, %)	-	-	1 (2.6%)	9 (40.9%)	1 (10%)	
Additional non-rheumatic disease * (n, %)	-	10 (12.3%)	4 (10.3%)	4 (18.2%)	2 (20%)	0.674
Family contact history of COVID-19 (n, %)	427 (84.4%)	67 (82.7%)	33 (84.6%)	20 (90.9%)	10 (100%)	0.603
Chest CT features of COVID-19 (n, %)	2 (0.4%)	10 (12.3%)	1 (2.6%)	4 (18.2%)	1 (10%)	<0.001
Outcome						
Symptomatic infection (n, %)	432 (85.4%)	76 (93.8%)	34 (87.2%)	19 (86.4%)	9 (90%)	0.345
Hospitalization (n, %)	10 (2%)	9 (11.1%)	-	2 (9.1%)	-	<0.001
Hospitalization duration ** (days) (median, min–max)	5.5 (4–30)	2 (1–21)	-	6 (5–7)	-	0.189
COVID-19 treatment						
HCQ (n, %)	-	4 (4.9%)	3 (7.7%)	9 (40.9%)	1 (10%)	
Antibiotic (n, %)	10 (2%)	5 (6.2%)	4 (10.3%)	-	1 (10%)	
Antiviral (n, %)	7 (1.4%)	19 (23.5%)	9 (23.1%)	8 (36.4%)	5 (50%)	
Anticoagulant (n, %)	3 (0.6%)	4 (4.9%)	-	2 (9.1%)	1 (10%)	

AIDs: Autoinflammatory diseases; AZT: Azathioprine; BD: Behçet disease; bDMARDs: biologic disease modifying antirheumatic drugs; BS: Blau syndrome; CAPS: cryopyrin associated periodic syndromes; cDMARDs: conventional disease modifying antirheumatic drugs; COVID-19: Coronavirus disease-2019; CRMO: Chronic recurrent multifocal osteomyelitis; CT: Computed tomography; CTD: Connective tissue disease; DADA2: Deficiency of Adenosine Deaminase 2; DM: Dermatomyositis; ERA: Enthesitis-related arthritis; FMF: Familial Mediterranean fever; GPA: Granulomatous polyangiitis; HCQ: Hydroxychloroquine; HIDS: Hyperimmunoglobulin D syndrome; HSP: Henoch-Schönlein purpura; JIA: Juvenile idiopathic arthritis; KD: Kawasaki disease; MMF: Mycophenolate mofetil; MTX: Methotrexate; oJIA: Oligoarticular juvenile idiopathic arthritis; PFAPA: Periodic fever, aphthous stomatitis, pharyngitis, and adenitis; pJIA: Polyarticular juvenile idiopathic arthritis; SD: Scleroderma; sJIA: Systemic juvenile idiopathic arthritis; SLE: Systemic lupus erythematosus; TA: Takayasu arteritis. * Healthy control group was not included in this analysis. ** Those who were not hospitalized were not included in the analysis.

**Table 2 jcm-11-02102-t002:** Demographic and underlying disease-related data comparisons between symptomatic and asymptomatic, and hospitalized and not hospitalized subjects.

	Symptomatic Infection			Hospitalization		
	Asymptomatic Group (n = 88)	Symptomatic Group (n = 570)	*p*	Hospitalized Group (n = 21)	Non-Hospitalized Group (n = 637)	*p*
Age (years) (median, min–max)	11.5 (0.33–20.47)	13.63 (0.4–20.95)	0.005	13 (0.5–19.68)	13.07 (0.33–20.95)	0.911
Gender			0.401			0.763
Female (n, %)	49 (55.7%)	290 (50.9%)		12 (57.1%)	327 (51.3%)	
Male (n, %)	39 (44.3%)	280 (49.1%)		9 (42.9%)	310 (48.7%)	
Disease			0.085			0.003
Healthy children (n, %)	74 (84.1%)	432 (75.8%)		11 (52.4%)	496 (77.9%)	
Patients with IRD (n, %)	14 (15.9%)	138 (24.2%)		10 (47.6%)	141 (22.1%)	
AIDs (n)	5	76		9	72	
JIA (n)	5	34		-	39	
CTD (n)	3	19		2	20	
Vasculitis (n)	1	9		-	10	
Follow-up duration * (months) (median, min–max)	54.5 (18–127)	64 (6–204)	0.063	65 (11–157)	62 (6–204)	0.825
Ongoing immunosuppressive treatments						
bDMARDs (n, %)	6 (6.8%)	39 (6.8%)	1	1 (4.8%)	44 (6.9%)	1
cDMARDs (n, %)	6 (6.8%)	32 (5.6%)	0.837	2 (9.5%)	36 (5.7%)	0.345
Non-rheumatic disease (n, %)	1 (1.1%)	19 (3.3%)	0.5	1 (4.8%)	19 (3%)	0.483
Family contact history of COVID-19 (n, %)	88 (100%)	469 (82.3%)	<0.001	19 (90.5%)	538 (84.5%)	0.757
Chest CT features of COVID-19 (n, %)	-	18 (3.2%)	0.152	9 (42.9%)	9 (1.4%)	<0.001

AIDs: Autoinflammatory diseases; bDMARDs: biologic disease modifying antirheumatic drugs; cDMARDs: conventional disease modifying antirheumatic drugs; COVID-19: Coronavirus diseases-2019; CT: Computerized tomography; CTD: Connective tissue disease; JIA: Juvenile idiopathic arthritis. * Healthy control group was not included in this analysis.

**Table 3 jcm-11-02102-t003:** Risk factor assessments for symptomatic infection and hospitalization due to COVID-19.

	Symptomatic Infection						Hospitalization		
	Model 1			Model 2					
	OR	95% CI	*p*	OR	95% CI	*p*	OR	95% CI	*p*
Age	1.046	1.003–1.091	0.037	1.046	1.003–1.092	0.035	0.977	0.897–1.063	0.585
Gender	0.817	0.517–1.291	0.387	0.818	0.517–1.294	0.391	1.128	0.458–2.778	0.794
bDMARD	0.307	0.176–1.730	0.307	0.569	0.179–1.911	0.361	0.230	0.028–1.863	0.168
cDMARD	0.159	0.139–1.380	0.159	0.545	0.109–2.735	0.461	0.704	0.141–3.522	0.669
IRD	2.452	1.014–5.929	0.047	-	-		5.785	2.179–15.363	<0.001
None	1			1			1	-	
AIDs	-	-		2.656	0.976–7.226	0.056	-	-	
JIA	-	-		2.062	0.434–9.789	0.362	-	-	
CTD	-	-		1.814	0.248–13.245	0.557	-	-	
Vasculitis	-	-		1.988	0.210–18.777	0.549	-	-	

AIDs: Autoinflammatory diseases; bDMARDs: Biologic disease modifying antirheumatic drugs; cDMARDs: Conventional disease modifying antirheumatic drugs; CTD: Connective tissue disease; IRD: Inflammatory rheumatic disease; JIA: Juvenile idiopathic arthritis.

## Data Availability

The data presented in this study are available on request from the corresponding author. The data are not publicly available due to local legal restrictions on data safety.

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
