# Peer review of "Comparisons of Clinical Features and Outcomes of COVID-19 between Patients with Pediatric Onset Inflammatory Rheumatic Diseases and Healthy Children"

_jcm, 2022, doi:10.3390/jcm11082102_

Round 1
Reviewer 1 Report
This interesting longitudinal study followed 658 ( 152 patiets with inflammatory rheumatic diseaese and 506 healthy children) who had been tested positive to SARS-CoV-2. The paper is overall well written and sound. I have some minor concerns:
- After reading and re-reading the paper I still struggle to understand how the participants were identified and followed. Was this a register linkage study or did you search the medical records? Did participants agree to participate? Could some patients have sought help at another hospital without it being registered in the study or that a hospital-consultation for an IRD is registered as a consultation in the study? How many were lost to follow-up?
- Were any participants immunised to SARS-CoV-2?
- There are some parts of the manuscript that need language editing: An example: Although the omicron variant of the virus, which currently dominates the outbreak, causes distinct clinical findings, considering that SARS-Cov-2 keeps going on to evolve rapidly and that we quite utilized the data regarding the previous coronaviruses to fight this novel one in the early days of the pandemic; we will always have numerous things to learn from the past.
Author Response
Response to Reviewer 1 Comments
This interesting longitudinal study followed 658 (152 patients with inflammatory rheumatic disease and 506 healthy children) who had been tested positive to SARS-CoV-2. The paper is overall well written and sound.
I have some minor concerns:
Point 1: After reading and re-reading the paper I still struggle to understand how the participants were identified and followed. Was this a register linkage study or did you search the medical records? Did participants agree to participate? Could some patients have sought help at another hospital without it being registered in the study or that a hospital-consultation for an IRD is registered as a consultation in the study? How many were lost to follow-up?
Response 1: Thanks to your valuable comments, we realized that we did not describe the patient selection in detail. At the beginning of the pandemic, we told our patients via phone or in face-to-face appointments that if they got the SARS-CoV-2 infection, they must inform us immediately. We routinely asked them their COVID-19 histories in the regular follow-ups. All our patients have our phone numbers, and we allow them to call us. Some of the infected ones were admitted to our hospital. We evaluated them and added their data to their medical records. Although some were admitted to another hospital, they informed us about their COVID-19 histories face-to-face at their regular follow-ups or by calling via phone. We asked about the clinical symptoms in detail and added their data to their medical records. However, the control group consisted of only healthy children admitted to our hospital who were SARS-CoV-2 PCR positive. All the clinical symptoms of all COVD-19 patients are noted in their medical records in our hospital. Moreover, COVID-19 data of all of the participants were verified by our national health registry with their permission. Then, data of all the participants were obtained from their medical records retrospectively. Yes, all the participants approved to be included in the study, and at least one of the participants' family members approved the written informed consent. None of the patients included in this study lost the follow-up. However, we acknowledge that some of our patients with IRD (particularly among the ones with milder disease course) may have not informed us about their positive PCR status, which may be a selection bias for our study (added this bias to the limitation paragraph) (Line: 327-328). We evaluated all the patients who declared to us that they were found to be SARS-CoV-2 PCR positive. Thanks to your great suggestions, we have tried our best to describe our patient selection method (Line: 69-82).
Point 2. Were any participants immunized to SARS-CoV-2?
Response 2: Vaccination data were available for only a few patients, and many of the patients were infected before the vaccination began worldwide. Therefore, we could not present this data, and we have defined the lack of vaccination data of the patients as a limitation of the study, thanks to your valuable comments (Line: 325-327).
Point 3. There are some parts of the manuscript that need language editing: An example: Although the omicron variant of the virus, which currently dominates the outbreak, causes distinct clinical findings, considering that SARS-Cov-2 keeps going on to evolve rapidly and that we quite utilized the data regarding the previous coronaviruses to fight this novel one in the early days of the pandemic; we will always have numerous things to learn from the past.
Response 3: Thanks to your great advice, we have re-written this part of the manuscript in a more clear way (Line: 233-237), and several minor corrections were also made-up.

Reviewer 2 Report
The Authors provided and interesting study on a very current topic in the pediatric field and in particular in the field of pediatric rheumatology.
Although with the limit of being a retrospective study it presented a good sample size and the limits of the study are well outlined at the end of the discussion. There are only a few minor comments. The highest hospitalization frequency and the highest risk for hospitalization was the IRD group, although the symptomatic infection frequency was not significantly different between IRD and healthy controls. How did the Author can explain this result? Could this result suggest a more cautious attitude towards SARS-COV-2 infection in this category of patients? What were the reasons that led to hospitalization in IRD patients?The other interesting results is that within the IRD groups, the highest hospitalization frequency regards patients not being treated with bDMARDS. This message is rather "reassuring" from the pediatric rheumatologist's point of view. Although looking at the baseline characteristics of the study population we can see that more than 90% of patients were on corticosteroid treatment and more than 50% of patients on cDMARDS were on MMF. The low number of hospitalized patients does not allow further analysis and, as a matter of fact, both bDMARDS and cDMARDS does not increase the risk of hospitalization (OR) but it could be an aspect to be considered in a future study on a major sample size.
Author Response
Response to Reviewer 2 Comments
Although with the limit of being a retrospective study it presented a good sample size, and the limits of the study are well outlined at the end of the discussion. There are only a few minor comments.
Point 1: The highest hospitalization frequency and the highest risk for hospitalization was the IRD group, although the symptomatic infection frequency was not significantly different between IRD and healthy controls. How did the Author can explain this result? Could this result suggest a more cautious attitude towards SARS-COV-2 infection in this category of patients? What were the reasons that led to hospitalization in IRD patients?
Response 1: Thanks to your magnificent contributions, we added the leading hospitalization causes of our IRD patients. Although there was no significant difference between the IRD patients and healthy children regarding their symptomatic infection frequencies, the presence of any IRD was found to increase the risk of symptomatic infection. We tried to mention this result in the text, thanks to your valuable comments (Line: 216-232).
Point 2: The other interesting results is that within the IRD groups, the highest hospitalization frequency regards patients not being treated with bDMARDS. This message is rather "reassuring" from the pediatric rheumatologist's point of view. Although looking at the baseline characteristics of the study population we can see that more than 90% of patients were on corticosteroid treatment and more than 50% of patients on cDMARDS were on MMF. The low number of hospitalized patients does not allow further analysis and, as a matter of fact, both bDMARDS and cDMARDS does not increase the risk of hospitalization (OR) but it could be an aspect to be considered in a future study on a major sample size.
Response 2: We are grateful for your great comments. These valuable sentences will inspire us for future studies.
